# Role of Metalloproteinases in Adhesion to Radicular Dentin: A Literature Review

**DOI:** 10.3390/ma17225674

**Published:** 2024-11-20

**Authors:** Marihana Valdez-Montoya, Mariana Melisa Avendaño-Félix, Julio César Basurto-Flores, Maricela Ramírez-Álvarez, María del Rosario Cázarez-Camacho, Miguel Ángel Casillas-Santana, Norma Verónica Zavala-Alonso, Seyla Nayjaá Sarmiento-Hernández, Erika de Lourdes Silva-Benítez, Jesús Eduardo Soto-Sainz

**Affiliations:** 1Maestría en Ciencias Odontológicas, Facultad de Estomatología, Universidad Autónoma de San Luis Potosí, San Luis Potosí 78290, Mexico; marihanavaldezm@gmail.com (M.V.-M.); nveroza@fest.uaslp.mx (N.V.Z.-A.); 2Maestría en Rehabilitación Oral Avanzada, Facultad de Odontología, Universidad Autónoma de Sinaloa, Sinaloa 80040, Mexico; marianaavendano@uas.edu.mx (M.M.A.-F.); juliobasurtof@uas.edu.mx (J.C.B.-F.); erikasilva@uas.edu.mx (E.d.L.S.-B.); 3Facultad de Odontología, Universidad Autónoma de Sinaloa, Sinaloa 80040, Mexico; dra.maricela_odontologia@uas.edu.mx (M.R.-Á.); mariacazarez@uas.edu.mx (M.d.R.C.-C.); 4Departamento de Ortodoncia, Facultad de Estomatología, Benemérita Universidad de Puebla, Puebla 72410, Mexico; miguel.casillas@correo.buap.mx; 5Maestría en Odontología Integral del Niño y el Adolescente, Facultad de Odontología, Universidad Autónoma de Sinaloa, Sinaloa 80040, Mexico; seylasarmiento@uas.edu.mx

**Keywords:** metalloproteinases, adhesion, radicular dentin, inhibitors, collagenolytic

## Abstract

Introduction: Root dentin is a porous and complex dental surface that may have irregularities and deposits of organic material. To achieve an effective bond between restorative materials and root dentin, it is necessary that the restorative materials adhere intimately to the dentin surface. Metalloproteinases (MMPs) are a group of proteolytic enzymes that perform an important role in degrading the extracellular matrix and remodeling connective tissue. The aim of this research was to determine the scientific evidence available on the role played by MMPs in adhesion to root dentin and their putative inhibitors. Materials and Methods: Several techniques have been used to evaluate the presence of MMPs in the root dentin of human and bovine teeth, such as Western blot, immunohistochemistry, immunofluorescence, and zymography, the latter also being used together with the EnzCheck assay to evaluate the inhibitory effect of adhesion protocols on the activity of root MMPs in vitro. Results: When analyzing the databases, 236 articles were found, 12 of which met the selection criteria. The variables analyzed were articles that evaluated different MMP inhibitors in root dentin. Conclusions: In the adhesion to radicular dentin, MMPs have a crucial role in the degradation of the extracellular matrix of dentin and the remodeling of the dentin surface because excessive MMP activity can be harmful to dental health, since excessive degradation of the extracellular matrix of dentin can weaken the tooth structure and decrease fracture resistance. Therefore, it is important to monitor MMP activity during root dentin bonding procedures.

## 1. Introduction

Dentin is a vital, elastic, permeable, and non-vascular tissue [1,2]. Its main functions are the protection of the pulp and absorption of the loads that the enamel receives [3,4]. Dentin is composed of approximately 55% mineral components, 30% organic components, and 15% water; however, these percentages vary in relation to the region of the tooth analyzed, the location of the tooth in the arch, and changes related to diseases or age [5]. There are marked differences between superficial and deep dentin in the coronal and root area. The number of dentinal tubules is lower in superficial dentin compared to the dentin close to the pulpar tissue. In addition, the tubular density is greater in the coronal portion of the tooth versus the apical portion [3]. Furthermore, in the coronal dentin, the diameter opening of dentinal tubules closes as they approach the dentinal junction [6], which results in a smaller quantity of collagen fibers, from the external dentin to the deep dentin [7].

Specifically, root dentin is a tissue with dentinal tubules, that extend from the pulp to the cementum. This characteristic feature confers permeability and moisture to this tissue [5]. Also, root dentin is a complex surface that may have irregularities and deposits of organic material, which can hinder effective bonding between the restorative materials and this tissue [8]. Therefore, it is necessary that the restorative materials adhere intimately to the dentin surface, in which several factors intervene in the adhesion of restorative materials, one of which is the presence of metalloproteinases (MMPs), a group of 28 modular endopeptidases that perform an important function in extracellular matrix remodeling and the control of extracellular signaling networks; they also regulate several processes, such as inflammation, bone growth, and angiogenesis, among others [9]. MMPs activity can drive several illness such as caries [10] since this process can trigger enzymatic endogenous MMPs’ activity, perpetuating this disease [11].

Interestingly, adhesive resin cementation is a moisture sensitive technique, even in teeth with root canal treatment. Therefore, pulpless dentin requires the same attention as the vital teeth during adhesive procedures [11]. Also, humidity control is decisive to achieve an effective and durable adhesion [12], with absolute isolation being the strategy to maintain an adequate and appropriate area for the adhesive protocol [13,14]. The above is mainly due to the action of MMPs, which are proteolytic enzymes capable of degrading the collagen fibers that remain unprotected after the incomplete infiltration of the monomers in the presence of moisture, allowing the progressive degradation of the hybrid layer [13,15], compromising the success of the adhesion of the restorative material to the dentinal substrate.

In addition, root dentin has less intertubular dentin that, in the presence of humidity, limits the adhesive potential of this tissue [8]. Therefore, it is important to inhibit MMP activity during dentin–root bonding procedures. To achieve this goal, it is important to have a broad knowledge of the nature of these endoproteinases and their possible implications on the dentin–root substrate.

For the above, the aim of this literature review was to identify the scientific evidence related to the role that matrix metalloproteinases play in adhesion to root dentin and to determine which agents have been reported to inhibit them.

## 2. Methods

The search strategy was executed in different databases such as the following: PubMed, Science Direct, Google Scholar, and Wiley, using the following MeSH and DeCS terms: metalloproteinase, root dentin, adhesives, root dentin, proteolysis, adhesive agent, matrix metalloproteinase, MMPs, collagenolytic activity, and MMPs inhibitor, in combination with the boolean operators AND, OR, and NOT.

Posteriorly, references were compiled with Mendeley, and duplicate articles were eliminated, obtaining 236 articles to which selection criteria were applied. Original articles published in indexed journals with in vitro evaluations of collagenolytic/gelatinolytic activity and MMPs inhibitors in dentinal substrate were included. Articles that did not evaluate root dentin as a restorative substrate were excluded, and articles with incomplete or not precise methodology were eliminated.

A total of 12 articles were finally analyzed ranging from January 2006 to June 2023.

## 3. Intraradicular Dentin as a Restorative Substrate

Dentin makes up the majority of the tooth structure, and its properties are crucial for restorative dentistry procedures [5]. It is composed of inorganic hydroxyapatite crystals (70% of the whole weight of the tissue) and several proteins (18%), mainly type I collagen (90%) and others (10%), and water (12%) [16]. Furthermore, dentin is produced by highly specialized and differentiated cells called odontoblasts [17]. Within the dentin, there are dentinal tubules that extend almost radially from the pulp towards the dentino–enamel junction and the cemento–dentinal junction. The diameter varies between 2 and 4 μm, with the number of tubules ranging from 18,000 to 21,000 per square millimeter, the number of which increases as they approach the dental pulp [18]. It is widely known that adhesion to dentin represents a greater complexity compared to enamel, since the contributions made by Nakabayashi in 1982 [19]. Also, differences may be found between the coronal and radicular dentin bond due to the different histological characteristics of the substrates and other variables, such as the high C-factor of the endodontic space, the presence of a smear layer, and the incompatibility between some adhesive systems and cements, as well as the limited access of the intraradicular space that can lead the clinician to different errors. In some cases, intraradicular dentin can be used as a restorative substrate in teeth that have been endodontically treated [13]. In addition, careful isolation and control of the working environment must be performed to avoid contamination and infection risks. It is also important to assess the patient’s periodontal health before deciding to use intraradicular dentin as a restorative substrate, as well as the function and aesthetics of the restored tooth. However, achieving a strong bond between the resin cement and the root dentin is a challenge, and, even if achieved, the bond degrades over time [20,21].

### 3.1. Resin–Dentin Bond

A smear layer is defined as “any residue of a calcified nature produced by the reduction or instrumentation of dentin, enamel, or cementum” [22]. On the other hand, a secondary smear layer is produced after the post space has been prepared and may contain the same material as the primary smear layer plus an aggregate of gutta-percha and cement residues that make its removal more difficult [23]. This removal is achieved by using etching acid or by performing different conditioning techniques [24]. Once removed, it is observed that the different adhesive systems increase their penetration depth within the dentinal tubules (between 10 and 80 μm) [25,26]. The penetrating action of the adhesive into the dentinal tubules creates “resin tags”. Regardless of the differences between coronal and radicular dentin mentioned above, the resin–dentin bonding interface is formed in the same way, through the infiltration of adhesive resin into the acid–demineralized dentin matrix, giving rise to the hybrid layer, which provides a mechanical interlocking bond strength [8].

Adhesive systems are a mixture of methacrylate-based resin monomers, with two cross-linking monomers or one end-polymerizable functional monomer, a photoinitiator system, organic solvents, and sometimes nanofillers. The hydrophilic functional monomers facilitate resin infiltration into the demineralized and damp tooth surface while the cross-linked hydrophobic resin monomers dispense stability, mechanical strength, and compatibility between the restorative resin and the adhesive system [27].

Currently, two strategies can be applied to resin bonding procedures: the etch and rinse (E&R) technique and the self-etch (SE) technique [28]. In E&R, to remove the smear layer, an acid is used; this provides a superficial layer of demineralized dentin 5 to 10 nm deep; then, the exposed mineral-free collagen network remains suspended in the rinse water and should be completely replaced by adhesive mixtures if a stable bond is to be achieved [29]. Opting for an SE system, which lacks the separate acid-etch step because the bonding comonomers at the same time demineralize and infiltrate the dentin substrate, thus reducing the discrepancy between the depth of the demineralized substrate and the infiltration depth achieved by the resin, results in a more homogeneous resin infiltration compared to E&R strategies. However, the effectiveness of SE bonding systems on enamel without separate acid-etch remains questionable [30,31]. In turn, these protocols have been simplified by reducing the application steps, giving rise to universal adhesives, which can be used in E&R or SE strategies with an extra chemical bonding agent. Nonetheless, it has been published that universal adhesives could not infiltrate the entire depth of the demineralized dentin created by the phosphoric acid used in the E&R systems [32]. Contrary to this, the hybrid layer of universal adhesives using the SE strategy appears to be more superficial and durable since these adhesives contain functional monomers capable of chemically interacting with hydroxyapatite and keeping collagen fibers protected after some time [32,33].

#### 3.1.1. Degradation of Dentin Bonding

The dentin–adhesive interface is susceptible to long-term natural degradation [34,35,36]. The durability of this interface is directly related to different variables such as the following: water absorption, masticatory forces, and proteolytic enzymes of dentin or from external origins, such as bacteria [37] and saliva, in addition to the intrinsic resistance of its components to degradation processes [38]. It is actually established that adhesive systems suffer the loss of their bond to dentin over time, and there is speculation that the degradation of the hybrid layer is related to the loss of bond strength [39,40]. In turn, the nature of the substrate influences adhesion since dentin is extremely difficult to adhere to, due to its moist and organic nature [41]. Usually, the bond strength reduces over a range of six months of aging, but it does not reach zero. Some bond strength is maintained even after long-term storage in water [39], which is involved with the hydrolysis of the collagen matrix of the hybrid layer combined with the degradation of the hydrophilic polymers of the adhesive systems [42]; maintaining or preserving the integrity of the collagen matrix is of utmost importance for adhesion to dentin over time [40].

#### 3.1.2. Hybrid Layer Degradation

Hydrolytic degradation only takes place in the presence of water. This chemical reaction has the ability to break covalent unions between polymers, resulting in loss of resin mass [43]. Therefore, in the hybrid layer, the principal reason for resin degradation is hydrolysis [44]. Dentin is a naturally moist substrate, and is, therefore, intrinsically hydrophilic; as a result, actual adhesives include mixtures of hydrophilic resin monomers, such as two-hydroxyethyl methacrylate (HEMA), in organic solvents and diluents, usually water, acetone, or ethanol. For adhesive infiltration in demineralized and moist dentin, the hydrophilic resin monomers are essential for producing a union between the adhesive and the substrate [45]. However, these hydrophilic resin monomers in adhesive formulations cause high water uptake via the resin systems, forming a hybrid layer after polymerization that could have a permeable membrane-like behavior, resulting in water movement across the bonded interface [46]. The hydrophilic phase of the adhesive is degraded by the water movement followed by the creation of large water-filled channels [47]. The above occurs since the water penetrates into the hydrophilic domains of the adhesive facilitating the leaching of the solubilized resin [44]. Then, the underlying insoluble collagen fibrils are exposed and vulnerable to proteolytic enzymes. [48]. Matrix proteases are hydrolases that add water through specific peptide bonds to break down the collagen “polymer” into “monomers”; however, the residual water interrupts this mechanism [47].

Another mechanism involved in the degradation of the hybrid layer is related to proteoglycans since these can organize binding water molecules, resulting in regulation of the affinity of collagen for water that can affect water replenishment during the formation of the hybrid layer [49]. Oral water/fluid absorption, polymer swelling, resin leaching, and hybrid layer degeneration could be caused by hydrolytic degradation activity of matrix metalloproteinases (MMPs) on the surface root collagen by dentin and are among the most essential mechanisms of bond deterioration [50,51,52].

## 4. Multiple Roles of Metalloproteinases Have Been Described over Time

Jerome Goss described the matrix metalloproteinases (MMPs) for the first time in 1962, and Charles Lupiere discovered that tadpole tail metamorphosis is calcium dependent. However, nowadays it is known that MMPs are zinc- and calcium-dependent, cell-secreting proteolytic enzymes [53,54]. These enzymes work at a neutral pH and cooperatively hydrolyze most of the proteins of the extracellular matrix (ECM), therefore leading to their degradation [55].

There have been over 20 MMPs identified in humans, and based on of substrate specificity, sequence similarity, and domain organization, they could be divided into the following groups [56]:(1)Collagenases (MMP-1, MMP-8, MMP-13, and MMP-18);(2)Gelatinases (MMP-2 and MMP -9);(3)Stromelysins (MMP-3, MMP-10, and MMP11);(4)Membrane-type matrix metalloproteinases (MMP-14, MMP-15, MMP-16, MMP-17, MMP-24, and MMP-25);(5)Others (MMP-12, MMP-19, MMP-20, MMP-21, MMP-22, MMP-23, MMP-27, and MMP-28).

Interestingly, these enzymes have an important role in several biological processes, such as embryogenesis, wound healing, normal tissue remodeling, and angiogenesis, as well as in diseases, such as cancer, arthritis, and tissue ulceration [57].

In an odontology context, matrix-MMPs are also involved in several physiological and diseased processes that occur in dentin, such as maturation in aging and formation and calcification of intertubular and intratubular dentin [58]. Also, MMPs participate in multiple processes, such as the progression of dentin caries and the degradation of the hybrid layer in restorations bonded with resin and dentin [59].

### 4.1. Implication of MMPs with the Performance of Composite Restoration

In the context of restorative materials, MMPs have been implicated in the longevity and clinical performance of composite restoration since these enzymes participate in bonding between the tooth and the composite restoration [60]. An appropriate hybrid layer formation is determinant of resin–dentin bonding through resin infiltration in demineralized dentin collagen, which couples adhesive/resin composites to the underlying mineralized dentin. Collagen fibrils in the dentin substrate can degrade via the activity of collagenolytic enzymes, resulting in decreased bond strength over time [60].

Nowadays, it is known that the bond strength at the dentin–adhesive interface is reduced via a normal hydrolytic degradation of dentin collagen. However, the inhibition of the expression of MMPs in the restorative substrate reduces the loss of bonding strength of the materials over time [61].

### 4.2. Presence of MMPs in Coronal Dentin

In coronal dentin, MMPs are produced by odontoblasts, fibroblasts, and immune cells present in the pulp tissue [62]. Their activity is regulated by various factors including inflammation [63], mechanical stress [64], and the presence of bacterial byproducts [65]. Also, MMPs participate in several physiological and pathological processes such as dentin formation and remodeling [66], dental caries development [67], dentin sensitivity [68], inflammatory responses during disease conditions like pulpitis [69], dentin repair and regeneration [70], and orthodontic tooth movement [71].

Specifically, MMPs reported to be present in the coronal dentin include MMP-2 (gelatinase-A) [72,73,74], MMP-9 (gelatinase-B) [74], and MMP-13 [75], as well as a poor concentration of MMP-1 (collagenase-1), MMP-3 (stromelysin-1), and MMP-8 (collagenase-2) [76]. These MMPs can be present in coronal dentin, regulating the organization and the mineralization of the collagen fibrils [77,78] and are involved in the process of caries progression [79,80]; and they also have an important role in the degradation of hybrid dentin layers [38,81].

Several drugs have been tested to evaluate the inhibition of host-derived MMPs from coronal dentin to improve the bond strength of coronal composite restorations, such as Indomethacin [60], 2% chlorhexidine, 0.3 M carbodiimide, and 0.1% riboflavin [81].

### 4.3. MMPs in Radicular Dentin

Fiber posts are used in the loss of substantial tooth structure. However, the longevity of fiber post restoration is mainly related to the adhesion between the radicular dentin and resin cement [5]. Any obstruction between these adhesive interfaces could result in a significant reduction in the longevity of the restorations [82]. Specifically, in root dentin, these enzymes can transform structural matrix proteins in signaling molecules; therefore, these enzymes perform a central role in dentin biomineralization and tissue regeneration therapies [83].

Several molecular techniques, such as immunostaining, liquid chip analysis, zymography, and Western blot, among others, have revealed that different MMPs are widely distributed in radicular dentin [52,84]. Such is the case of MMP-2 and -8, which are considerably distributed in root dentin, while MMP-3 shows a greater presence in middle and apical third of the root [84]. Interestingly, other reports reveal that MMP-2 can be found in the demineralized root dentin matrix while MMP-9 is mainly present in the mineralized compartment of dentin in a minor amount, whereas MMP-8 can be evenly distributed in crown and root dentin [52]. It is worth noting that MMP-20 has been found but in a smaller amount in root dentin, with MMP-2 and MMP-8 being the main MMPs reported in this substrate [83]. Another MMP that has been identified as specific to root dentin has been MMP-13, which is not found in coronal dentin, and which has interestingly been determined to modify its expression as the cariogenic process progresses [75]. All MMPs detected in radicular dentin are summarized in Table 1.

## 5. Synthetic and Natural MMP Inhibitors

Contemporary advances enhance the resistance of collagen fibrils against enzymatic damage and inactivating proteinase activities, with the application of collagen cross-linking agents and MMP inhibitors [27,88]. These inhibitors have been implemented as another possibility to control the activity of endogenous MMPs and improve the stability of the hybrid layer. Among the reported MMP inhibitors (synthetic and natural) are chlorhexidine, ethylenediaminetetraacetic acid, benzalkonium chloride, galardin, green tea extract, riboflavin, and baicalein [81].

### 5.1. Clorhexidine

A potential candidate to perform the task of inhibiting MMPs is chlorhexidine (CHX), which has demonstrated an inhibitory effect on gelatinases (MMP-2 and MMP-9) [89] and collagenases (MMP-1 and MMP-8) [90]. Interestingly, a study by Hebling et al., in 2005, showed that the application of CHX prior to bonding preserves collagen integrity for at least 6 months [38]. Also, Carrilho et al., in 2007, demonstrated that this solution has no effect on the bond strength and morphological aspects of hybrid layers and promotes the complementary use of CHX in acid-etched dentin to delay the degradation of the hybrid layer [91]. CHX has been shown to inhibit another class of collagen-degrading enzymes (cysteine cathepsins) which are also present in dentin [15,92].

### 5.2. Carbodiimides

Carbodiimides (EDCs) are agents that directly conjugate carboxylates to primary amines without changing the final cross-link part (amide bond) between the target molecules [93]. Proteases that bind to the demineralized dentin matrix are inactivated by carbodiimide, which causes a change in the three-dimensional conformation of the MMPs, inactivating the catalytic sites [35]. This solution does not contain toxic components with a high biocompatibility, having been fully removed, which improves the mechanical properties [91]. Mechanically, carbodiimide acts as a collagen cross-linker; in this process, proteolytic enzymes are structurally modified, preventing the deterioration of the created bonds [35].

### 5.3. Epigallocatechin Gallate

Epigallocatechin gallate (EGCG), one of the main polyphenols in green tea, provides numerous functions, such as anti-inflammatory, antimicrobial, anti-collagenolytic, antioxidant, and anticancer effects [94]; this cross-linker can stabilize the collagen chain [95]. In addition, it can decrease collagen biodegradation, increasing the number of collagen cross-links via the interaction of hydrogen molecules of galloyl groups [96].

### 5.4. Baicalein

Baicalein (BAI) is one of the major flavonoids from *Scutellaria baicalensis*. It is a natural plant polyphenol that shares a molecular structure resembling phenolic hydroxyl functional groups, suggesting that they share similar dentinal cross-linking properties [97].

Previous studies have determined that some of their hydroxyl groups present a potent chelation capacity for several metals such as Zn. It has been described that MMPs are Zn/Ca-dependent enzymes, so this could be the mechanism through which the BAI can inhibit MMPs [98,99]. Also, BAI could cross-link and modify the three-dimensional structure or molecular mobility of MMPs, leading to the loss of the latter’s collagen enzymolysis capacity [100].

## 6. Adhesive Systems with Greater Effectiveness When Used with MMP Inhibitors

The literature shows mixed results regarding adhesive systems used in studies using the radicular dentin substrate to bond the restoration. Self-etch adhesives have been reported to promote collagenolytic activity in intraradicular dentin. Instrumented radicular dentin presents an implicit collagenolytic activity; it has been reported that this is activated by mild self-etch (SE) adhesives [101,102]. In contrast to phosphoric acid, the mildly acidic resin monomers do not have the capacity to denature activated enzymes. As a consequence, partial self-degradation of an incompletely demineralized collagen matrix via hydrolysis may occur in the presence of water [103].

### 6.1. Effect of MMP Inhibitors When Included in Root Dentin Adhesion Protocols

#### 6.1.1. Carbodiimide Reduces Endogenous Enzymatic Activities Within the Hybrid Layer

Comba et al., in 2019, proved that two-step E&R adhesives caused a significantly higher increase in collagenolytic activity compared to three-step E&R systems, through in situ zymography. They clearly show an important inhibition of MMP-induced gelatinolytic activity within the hybrid layer after the colocation of EDC as an inhibitor, regardless of the type of adhesive (two- and three-step E&R), concluding that EDC demonstrates that inside of the intraradicular dentin layer exists a reduction in the endogenous enzymatic activities [35].

#### 6.1.2. Green Tea Extracts Demonstrate Inhibitory Effect on MMPs

Recently, the inhibition of MMPs by epigallocatechin gallate, a green tea-derived polyphenol, has been demonstrated, as well as the modification of Single Bond 2 (E&R system) adhesive with epigallocatechin gallate in terms of their bonding stability to intraradicular dentin. Both epigallocatechin gallate and epigallocatechin-3-O-(3-O-methyl) gallate inhibits root dentin-derived MMPs depending on the concentration, and the inhibitory activity of epigallocatechin-3-O-(3-O-methyl) gallate was stronger than that of epigallocatechin gallate at the same concentration. The adhesive, modified with these two methylated and unmethylated polyphenols, had a higher ejection force than Single Bond 2 after thermocycling, showing no correlation with concentration, and its ejection force was not compromised by this modification [84]. Furthermore, application of a 2% green tea extract, after a 15 s conditioning with 37% phosphoric acid, showed that the etch-and-rinse system increased bond durability to dentin after six months [104].

#### 6.1.3. Baicalein Inhibits Dentinal Gelatinase and Collagenase Activities

Interestingly, it has been demonstrated that BAI at a concentration of 0 to 5.0 ug/mL did not affect the conversion of adhesives. However, BAI at a concentration of 2.5 ug/mL inhibits dentin binding with gelatinase and collagenase activity; this produces an augmentation of the microtension bond strength and a decrease in nanoleakage in vitro, showing that BAI used as a pre-conditioner in an E&R adhesive system exhibited an anti-MMP function and effectively improved the durability of the resin–dentin bond in vitro [100].

#### 6.1.4. Chlorhexidine Is an MMP Inhibitor Substance

Multiple studies have tested CHX, verifying the inhibitory action of this substance on MMPs; such is the case of Jianfeng et al. who observed that, after 18 months, there was an important reduction in the bond strength of all groups (Control ED Primer without CHX, ED Primer + 0.5% CHX, and ED Primer + 1% CHX). The reduction in bond strength in the 1% CHX group was significantly lower than that of the control group and the 0.5% CHX group. Incorporating 1% CHX into the adhesive protocol can prolong the longevity of the bond in root dentin [105].

Finally, a study that included two groups (a control group: Single Bond + storage in artificial saliva without application of an MMP inhibitor and another group: Single Bond + storage in artificial saliva with 2% CHX) showed that pretreatment with CHX has no immediate effect on bond strength in vitro. However, after 6 months, an effect is found. Interestingly, when the sample is stored in artificial saliva plus protease inhibitors, the bond strength was not modified compared to those stored in artificial saliva without protease inhibitors, resulting in the use of CHX at 2% after etching acid and before adhesive application being a common step in oral rehabilitation, since in vitro studies have shown that the use of CHX decreases the degradation of the hybrid layer compared to when it is not used [106].

All the adhesion protocols employed to inhibit the effect of radicular MMPs are summarized in Table 2.

## 7. Discussion

The first report of collagenolytic activity in dentin was on 1983 by Dayan et al. Ref. [107] and this issue was clarified by Tjäderhane et al. that asserted this capacity to matrix metalloproteinases [79]. However, Pashley et al. exhibited that collagen could be reduced with time under aseptic conditions, via the action of intrinsic matrix proteases. Since then, research has been focused on determining the types, location, importance, and implications of the enzymes related to the dentin extracellular matrix, as well as elucidating the strategies to inhibit them. It has been shown that there are MMPs in coronal dentin, such as MMP-2, MMP-8, and MMP-9, as shown via functional and immunological assays. Through this search for information, it was concluded that the MMPs present in root dentin are MMP-2, MMP-3, MMP-8, MMP-9, and MMP-13 [108].

The composition of dentin can vary depending on the area, the approach to pulp tissue, and if the matrix is decayed or demineralized [109]. These characteristics could influence the mechanical features of dentin and the successful bonding of different materials to this substrate. The MMPs’ distribution depends on their functions during root dentin formation as well as to the disease stage or physiological conditions following the eruption of permanent teeth [84]. In the search for less invasive and conservative treatments, every possibility is exhausted, without knowing the implications that these actions may cause, such as the controversy over the use of affected dentin in bonded restorations, which was described by Toledano et al. in 2010. In their study, they observed that the expression of MMPs was lower in healthy dentin and more intense in caries-infected dentin. However, caries-affected dentin showed intermediate immunoreactivity. They also determined that a high expression of MMP-2 in decay dentin (radicular and coronal) compared to healthy dentin (radicular and coronal) may imply a faster degradation of the hybrid layer when decay dentin is used as a substrate for restorations, which means that it is important to completely rule out both infected and affected root dentin as a restorative substrate [85].

Another question that arises is whether there is scientific evidence to support the intervention of exogenous MMPs, such as salivary ones; however, the literature shows contradictory results on this point [110]; although, it is possible for salivary MMPs to assault resin–dentin bonds [111]. It has been reported that incubation of resin–dentin bonds with exogenous collagenase or cholesterol esterase and collagenase have no additional effect on the bond strength on the inhibition in the control groups without exo-collagenases [112]. The above findings suggest that enzymes found in the saliva may be too big to go inside resin–dentin bonds [113], ruling out the involvement of exogenous proteolytic enzymes in this degradation process.

During the dentin bonding process, MMPs are liberated through acid etching and are also activated via adhesive application, exposing collagen fibrils inside the hybrid layer [27,103]. Also, etching of intraradicular dentin exposes the catalytic domains of MMPs which bind to the collagen matrix in radicular dentin, activating the MMPs’ precursors [114]. MMP inhibitors have been shown to preserve binding strength over time by altering the arrangement of the catalytic domains or allosterically inhibiting other modular domains involved in collagen degradation [35,114].

Furthermore, soft self-etching adhesives create hybrid layers, and despite the simultaneous nature of these adhesive etching and self-etching systems, these are not perfect as they contain nanovoids that are permeable to water. MMPs are hydrolase-type enzymes, which hydrolyze peptide bonds in collagen molecules with the help of water. Simplified one-step self-etching adhesives have been used, which are very prone to absorbing water, allowing water to easily penetrate these layers and allowing the MMPs to exert their hydroactive action, causing significant damage to the longevity of bonded intraradicular restorations [101].

Studies using failure mode analysis suggest that EDC inhibits MMPs. Interestingly, when dentin is pretreated with acid-etching plus EDC, almost the same number of adhesion failures occur immediately as well as 12 months after treatment, using artificial saliva in an in vitro study. The above finding is accredited to the cross-linking MMPs in preserving the integrity of the collagen network. Despite this, bond strength decreases in the middle and apical regions of the posterior tooth space, possibly because the effectiveness of EDC decreases as collagen decreases in the apical region [35].

Furthermore, tissue inhibitors of matrix metalloproteinases (TIMPs) are a potentially important topic to develop, since these enzymes are often present in extracellular matrices to regulate MMP activities [80,115], showing a significant reduction after pulp tissue removal and canal filling with synthetic materials through endodontic therapy; in addition, in vitro, a low cytotoxicity reaction has been reported; although, so far no removal problems of the inhibitors have been reported [116,117,118]. Therefore, this is another important reason to consider using synthetic MMP inhibitors to prevent the degradation of attached intraradicular dentin.

## 8. Future Directions

Interestingly, the presence of MMP-3 has been reported in root dentin [84]. However, it has not been evaluated whether chlorhexidine has an inhibitory effect in vitro; so, it could be interesting to perform both in vitro and in vivo studies.

It has been reported that carbodiimide [35], green tea extracts [84,104], and balcalein [100] exhibit an inhibitory effect on endogenous MMPs in root dentin. However, these studies are limited to in vitro experiments, so the evaluation of these effects observed in the laboratory must be verified on in vivo studies and then transferred to clinical studies, which will allow for the verification of low toxicity on the tissues surrounding the tooth, in addition to the positive effects that these agents have on MMPs. The above is necessary because, without this in-depth research, it will be difficult to have the commercial availability of these products. However, currently, there is a universal bond system that includes CHX (0.2%) in its formula [119].

Remarkably, the initial bond degradation phenomenon often goes undetected radiographically, leading clinicians to mistakenly assume that root canal adhesive bonds remain effective over time. This underscores the need for further in vivo and in vitro testing to assess the longevity of root canal treatments and restorations.

Also, future research should consider studies on intertubular fluids and their ability to inhibit the infiltration of adhesive monomers, using confocal laser scanning microscopy and analyzing their impact on hybrid layer degradation. It should also consider the comparison of different adhesive systems, new biofunctionalized adhesives that may have the ability to inhibit enzymatic action, and how clinical steps may influence MMP activation.

## 9. Conclusions

The distribution and concentration of MMPs vary significantly between coronal and root dentin. Interestingly, MMP-2, -3, -8, -9, and -13 are the most reported in greater proportion in root dentin and have been described as being inhibited by agents, such as chlorhexidine, carbodiimide, balcalein, and epigallocatechin gallate, enhancing the durability of adhesive bonds to root dentin. This phenomenon has been noted both when these agents were employed as a standalone treatment and when integrated into the composition adhesive system. The results of the present review support the benefits of pretreatment with carbodiimide, chlorhexidine, epigallocatechin gallate, and balcalein on acid-etched root dentin. Interestingly, despite requiring an extra minute of clinical chair time, this procedure yields beneficial results by preventing bond degradation in root dentin through MMP cross-linking.

CHX has been shown to be effective in inhibiting collagen degradation caused by naturally occurring host-produced MMPs, including gelatinases (MMP-2 and MMP-9) and collagenases (MMP-8 and MMP-13), but to date, no studies have been performed to demonstrate this inhibitory efficacy of chlorhexidine on stromelysins (MMP-3). Despite this, CHX is the only inhibitor that is effective on most root dentin endoproteinases, and the other MMP inhibitors mentioned in this review are not currently available for clinical use because more studies are still needed to support their effectiveness and, above all, their safety in clinical stomatological use, which supports CHX as the best MMP inhibitor that currently exists.

## Figures and Tables

**Table 1 materials-17-05674-t001:** Studies that show the presence of MMPs in radicular dentin using *in vitro* analysis. (N/D: not data).

Reference	*N*	H = Human, B = Bovine	Age Ranges (y = years, M = months)	Technique	Results
[52]	40	H	20–30 y	Zymography	Gelatinolytic activity via MMP-2 and MMP-9 was found in coronal and radicular dentin. Specifically, in radicular dentin, MMP-2 was more notorious in demineralized dentin and MMP-9 in mineralized dentin and presented lower levels in general. MMP-8 was found equally distributed in coronal and root dentin.
[85]	10	H	18–30 y	IF	Caries stimulates the expression of MMP-2 in healthy, caries-affected, and caries-infected dentin, both coronal and radicular. Caries-affected dentin showed a lower intensity of MMP-2 expression than infected dentin, but greater immunoreactivity than healthy dentin. Similar features were observed in coronal and radicular dentin.
[86]	30, 40	H, B	18–25 y, 24–36 M	Zymography	MMP-2 and -9 were observed in coronary and radicular dentin of bovine and human teeth. Bovine dentin was found to be a reliable substrate for studies that involve MMP-2 and -9 activity.
[75]	7	H	N/D	Western blot	MMP-13 was found in radicular dentin with different expression as caries progressed; however, in the coronal dentin group, it was not expressed.
[87]	20	H	18–31 y	Zymography	The MMP-2 enzyme from human coronal and radicular dentin is influenced by pH: at a low pH, the enzyme is in a latent form; however, when the pH is close to neutral, collagen degradation by the matrix-bound enzyme is found.
[84]	106	H	20–30 y	IHC and IF	MMP-2 and MMP-8 are distributed in the radicular dentin, while MMP-3 exhibits a higher concentration in the middle and apical third of the root.

**Table 2 materials-17-05674-t002:** Inhibitory effect on radicular MMPs using adhesion protocols in vitro. (N/A: not applicable, RD: radicular dentin).

Reference	*N*	Bond Technique	No. Steps	Adhesive System	Substratum	Groups/MMPs Inhibitor	Inhibitor Action Time (m = Minutes, M = Months)	Technique	Results
[101]	50	N/A	N/A	N/A	RD	Group 1: Control +	N/A	EnzCheck	Instrumented intraradicular dentin showed latent collagenolytic activity that was activated by mild self-etching adhesives, with CHX being the treatment with the most favorable results in all the groups.
		N/A	N/A	N/A		Group 2: Control −/CHX 2%	1 m		
		N/A	N/A	N/A		Group 3: Control −/EDTA 17%	1 m		
		HE	2	Clearfil Liner Bond 2V		Group 4: Primer + CLB2	1 m		
		HE	1	Clearfil Tri-S Bond		Group 5: CTriS	1 m		
		HE	2	Clearfil Liner Bond 2V		Group 6: CHX 2% + CLB2	10 m		
		HE	1	Clearfil Tri-S Bond		Group 7: CHX 2% + CTriS	10 m		
		HE	2	Clearfil Liner Bond 2V		Group 8: CLB2+CHX 2%	10 m		
		HE	1	Clearfil Tri-S Bond		Group 9: CTriS+CHX 2%	10 m		
[35]	20	E&R	3	All Bond 3	RD	Group 1: AB3	N/A	Zymography	Application of XPB adhesive (2sE&R) resulted in significantly higher gelatinolytic activity compared to AB3 (3sE&R). No significant influence was identified. The use of EDC notably improved the fiber post bond strength at one year. Also, application of 0.3 M EDC before bonding significantly reduced gelatinolytic activities inside root hybrid layers, and EDC was effective in preserving fiber post FU over time by reducing the activities of endogenous intraradicular proteases.
		E&R	3	All Bond 3		Group 2: 0.3M EDC + AB3	1 m		
		E&R	2	Prime and Bond XP		Group 3: XPB	N/A		
		E&R	2	Prime and Bond XP		Group 4: 0.3M EDC + XPB	1 m		
[84]	80	N/A	N/A	N/A	RD	Group 1: Control (Deionized Water)	1 m	EnzCheck	MMP-2 and MMP-8 are commonly distributed in root dentin, whereas MMP-3 present higher fluorescence intensity in the middle and apical third of the root. Moreover, MMP-2 is more present in each third of the tooth root compared with the content of MMP-3 and MMP-8. The MMP inhibitory activity of EGCG-3ME was stronger than EGCG at the same concentration. The inhibitory effect stabilizes by the first 8 h, and after 48 h, the inhibitory activity decreased in a concentration dependent manner.
						Group 2: Control + (1,10-phenanthroline)	1 m		
						Group 3: 200 μg/mL EGCG (E200)	1 m		
						Group 4: 400 μg/mL EGCG (E400)	1 m		
						Group 5: 600 μg/mL EGCG (E600)	1 m		
						Group 6: 200 μg/mL EGCG-3Me (E− 3Me200)	1 m		
						Group 7: 400 μg/mL EGCG-3Me (E− 3Me400)	1 m		
						Group 8: 600 μg/mL EGCG-3Me (E− 3Me600)	1 m		
[105]	30	HE	1	ED Primer	RD	Group 1: ED control first Without CHX	N/A	N/A	At 18 months, a significant reduction in bond strength of all groups remains. The CHX at 1.0% group exhibited the significantly less reduction in comparison to the groups of CHX 0.5% and the control, concluding that incorporating CHX 1.0% in the ED primer can prolong the longevity of the bond in root dentin.
		HE	1			Group 2: ED primer + 0.5% CHX	1 m		
		HE	1			Group 3: ED primer + 1.0% CHX	1 m		
[104]	30	E&R	2	Adapter Single Bond 2	R.D., C.D.	Group 1: TV 2%	1 m	N/A	In the μTBS test, for all groups, there was no significant difference after 24 h. After 6 months, the TV group had higher microtensile values. Applying 2% green tea extract increased the durability of the bond in the E&R system. CHX and the control had no effect on bond strength after water storage.
		E&R	2	Adapter Single Bond 2		Group 2: CHX 2%	1 m		
		E&R	2	Adapter Single Bond 2		Group 3: Control	N/A		
[100]	N/E	E&R	2	Adapter Single Bond 2	R.D., C.D.	Group 1: ASB2 + BAI 0.1 ug/mL	2 m	EnzCheck	BAI at a concentration of 0 to 5.0 µg/mL did not affect the adhesive conversion. Although, it did inhibit gelatinase and collagenase activities at a dose of 2.5 µg/mL, increasing microtensile bonding force and decreasing nanoleakage in vitro. BAI used as a preconditioner in a Syst. E&R adhesive has an anti-MMP function and effectively improves the durability of resin–dentin bonding in vitro, which has potential value in clinical bonding procedures.
		E&R	2	Adapter Single Bond 2		Group 2: ASB2 + BAI 0.5 ug/mL	2 m		
		E&R	2	Adapter Single Bond 2		Group 3: ASB2 + BAI 2.5 ug/mL	2 m		
		E&R	2	Adapter Single Bond 2		Group 4: ASB2 + BAI 5.0 ug/mL	2 m		
		E&R	2	Adapter Single Bond 2		Group 5: ASB2 + CHX 2%	2 m		
		E&R	2	Adapter Single Bond 2		Group 6: ASB2 + DMSO 1%	2 m		
		E&R	2	Adapter Single Bond 2		Group 7: ASB2 control + distilled water	2 m		
[106]	14	E&R	2	Single Bond	R.D., C.D.	Group 1: CHX 2%	6 M	N/A	CHX pretreatment did not affect in vitro bond strength at the immediate testing period. Storage of 6 months resulted in a significant reduction in the bond strength of the CHX and control groups. Storage in artificial saliva without protease inhibitors minimized the binding strength in the control group. In the CHX group, the decrease was 23.4%. The remaining adhesive strength was elevated in the CHX group. In vitro preservation in artificial saliva with protease inhibitors did not affect binding strength compared with the storage in artificial saliva without protease inhibitors.
		E&R	2	Single Bond		Group 2: Control	N/A		

## Data Availability

Not applicable.

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
