# Peer review of "Role of Metalloproteinases in Adhesion to Radicular Dentin: A Literature Review"

_materials, 2024, doi:10.3390/ma17225674_

Round 1

Reviewer 1 Report

Comments and Suggestions for Authors

Dear Author,

The article entitled "Role of Metalloproteinases in Adhesion to Radicular Dentin: A Literature Review" explores the role of metalloproteinases (MMPs) in root dentin adhesion and the mechanisms of inhibition of these enzymes to improve the durability of adhesive restorations. The main objective of the review was to present the available scientific evidence on how MMPs, which are proteolytic enzymes, affect the integrity of dentin tissue during the adhesion process and to identify agents that can inhibit their activity. The article is well structured regarding the literature review and details the role of MMPs in dentin adhesion. The discussion and conclusion are well-developed, with a clear perspective on the need for further clinical studies.

However, the article has some limitations regarding the methodology used to select the reviewed articles. First, it does not mention the specific databases used, such as PubMed or Scopus, nor does it detail the period of selection of the studies, that is, the years in which the research was collected. This lack of clarity may compromise the rigor of the literature review since transparency and replicability are fundamental in any scientific study.

In addition, although the article states that 12 articles were selected from 236 found, it does not specify the search terms used or the inclusion and exclusion criteria. This information must be present to ensure the reliability of the research is maintained.

Therefore, I suggest the article include a more detailed "Methods" section that clearly explains the process of searching, selecting, and analyzing the reviewed studies. This would not only increase the transparency of the work but also reinforce its credibility.

Author Response

REVIEWER 1

Dear reviewer we appreciated your observations and suggestions to improve this manuscript.

Comments and Suggestions for Authors

Dear Author,

The article entitled "Role of Metalloproteinases in Adhesion to Radicular Dentin: A Literature Review" explores the role of metalloproteinases (MMPs) in root dentin adhesion and the mechanisms of inhibition of these enzymes to improve the durability of adhesive restorations. The main objective of the review was to present the available scientific evidence on how MMPs, which are proteolytic enzymes, affect the integrity of dentin tissue during the adhesion process and to identify agents that can inhibit their activity. The article is well structured regarding the literature review and details the role of MMPs in dentin adhesion. The discussion and conclusion are well-developed, with a clear perspective on the need for further clinical studies.

However, the article has some limitations regarding the methodology used to select the reviewed articles. First, it does not mention the specific databases used, such as PubMed or Scopus, nor does it detail the period of selection of the studies, that is, the years in which the research was collected. This lack of clarity may compromise the rigor of the literature review since transparency and replicability are fundamental in any scientific study.

In addition, although the article states that 12 articles were selected from 236 found, it does not specify the search terms used or the inclusion and exclusion criteria. This information must be present to ensure the reliability of the research is maintained.

Therefore, I suggest the article include a more detailed "Methods" section that clearly explains the process of searching, selecting, and analyzing the reviewed studies. This would not only increase the transparency of the work but also reinforce its credibility.

RESPONSE: We appreciated your suggestion; we attend these comments. A “Methods” section underlined in green was included with the specific databases used, period of selection of the studies, the inclusion and exclusion criteria.

Reviewer 2 Report

Comments and Suggestions for Authors

A very detailed review for the role of MMPs in dentin bond degradation.

However it seems that this review should point some issues:

1) Since most of the the studies about MMPs inhibitors are in vitro, what is derived from clinical studies? Is it easy to organize a clinical study with this background?

2) If CHX is effective, how safe could be to adapt an aditional step in clinical practice (before adhesive bonding agents?)

3) Are all MMP inhibitors safe to use and are they water soluble enough to be removed before bonding agent or they do not cause any effect if  not thoroughly rinsed?

4) Are there commercial products (bonding agents,Resin cements,universal adhesives) that include MMPs?

Author Response

REVIEWER 2

Dear reviewer we appreciated your observations and suggestions to improve this manuscript.

Comments and Suggestions for Authors

A very detailed review for the role of MMPs in dentin bond degradation.

However it seems that this review should point some issues:

  • Since most of the studies about MMPs inhibitors are in vitro, what is derived from clinical studies?

RESPONSE: We appreciated your observation. Until now, there are no clinical studies that evaluated MMP inhibitors. This information was included in future directions underlined in green.

Is it easy to organize a clinical study with this background?

RESPONSE: We appreciated your comment. We considered it might not be easy to organize a clinical study. Mainly for the multiple variables that could modified results. Like the pH of the oral environment, the amount of cross-linking collagen, the type of filler particles in the adhesive and cement, the concentration of water in the primer or adhesive, the presence of residual water, the amount of residual tooth structure, partial removal of endodontic sealant cement, vertical or horizontal loads on the teeth and patient parafunctions to mean a few. All of these are constant concerns of many researchers that make difficult to carry out this type of studies.

  • If CHX is effective, how safe could be to adapt an aditional step in clinical practice (before adhesive bonding agents?)

RESPONSE: We appreciated your observation. Actually, the use of CHX after etching acid and before of adhesive applied it’s a common step in oral rehabilitation, since, in vitro studies have shown that the use of CHX decreases the degradation of the hybrid layer compared to when it is not used. This information is included in the line 425.

  • Are all MMP inhibitors safe to use

RESPONSE: We appreciated your observation, all MMP inhibitors mentioned in this review have been reported safe to use, showing low cytotoxicity in vitro. MMP inhibitors are not placed in direct contact with cells and the exposure times to dentin are low. This information was included in discussion underlined in green.

and are they water soluble enough to be removed before bonding agent or they do not cause any effect if not thoroughly rinsed?

RESPONSE: Until now, have not been reported removal problems on in vitro analyses. In fact, it has been shown increased dentin mechanical properties in vitro. This information was included in discussion underlined in green.

4) Are there commercial products (bonding agents,Resin cements,universal adhesives) that include MMPs?

RESPONSE: We appreciated your observation, until now does not exist on the market bonding agents, resin cements or universal adhesives that include MMPs. In the case of MMP inhibitor’s actually Ultradent have a universal bond system that include CHX (0.2%) in its formula. This information was included in future directions underlined in green.

Round 2

Reviewer 1 Report

Comments and Suggestions for Authors

Dear Authors,

After thoroughly reading the manuscript, including the revisions suggested by the reviewers, I consider the revised version suitable for publication in the journal Materials. All modifications have been implemented, making the text more precise and informative.

Kind regards